# Youth Engagement in Climate Change Action: Case Study on Indigenous Youth at COP24

**Makenzie MacKay [1],\*, Brenda Parlee [1] and Carrie Karsgaard [2]** 

[1]  Department of Resource Economics & Environmental Sociology, University of Alberta, Edmonton, AB T6G 2H1, Canada; bparlee@ualberta.ca
[2]  Department of Educational Policy Studies, University of Alberta, Edmonton, AB T6G 2G5, Canada; karsgaar@ualberta.ca
\*  Correspondence: mmackay2@ualberta.ca

**Abstract:** While there are many studies about the environmental impacts of climate change in the Canadian north, the role of Indigenous youth in climate governance has been a lesser focus of inquiry. A popularized assumption in some literature is that youth have little to contribute to discussions on climate change and other aspects of land and resource management; such downplay of youth expertise and engagement may be contributing to climate anxiety (e.g., feelings of hopelessness), particularly in remote communities. Creating opportunities for youth to have a voice in global forums such as the United Nations Conference of Parties (COP24) on Climate Change may offset such anxiety. Building on previous research related to climate action, and the well-being of Indigenous youth, this paper shares the outcomes of research with Indigenous youth (along with family and teachers) from the Mackenzie River Basin who attended COP24 to determine the value of their experience. Key questions guiding these interviews included: How did youth impact others? and How did youth benefit from the experience? Key insights related to the value of a global experience; multiple youth presentations at COP24 were heard by hundreds of people who sought to learn more from youth about their experience of climate change. Additional insights were gathered about the importance of family and community (i.e., webs of support); social networks were seen as key to the success of youth who participated in the event and contributed to youth learning and leadership development.

**Keywords:** youth; climate change; activism; Indigenous; leadership; learning; networks; positive development; webs of support

## 1. Introduction

Climate change is having significant impacts on Indigenous peoples of the territorial and provincial norths of Canada [1,2]; warming temperatures, extreme weather events, and changes in valued natural resources may be contributing to anxiety among First Nations and Inuit peoples, including youth who are already characterized as socio-economically vulnerable as a result of colonial institutions and policies such as residential school programs [3–8]. Research elsewhere has revealed how climate discourse in schools and from the media can create or amplify fear, anxiety, and hopelessness among children and youth [8–10]. There is also important work being done to understand youth resilience to climate change and other stresses [11–13]; within this body of work are emerging insights about the role of youth in natural resource management and climate governance [14–17]. However, little of this work has looked closely at the experience of Indigenous youth.

This paper explores the value of Indigenous youth engagement in climate governance as an opportunity to offset the anxieties related to climate change. Drawing on interviews with youth from First Nations and Inuvialuit communities in the Northwest Territories, Yukon, and Alberta, Canada

who attended the 2018 Conference of the Parties on Climate Change in Katowice, Poland (COP24), the paper shares narratives about the value of global experiences and the importance of family and community supports. Specifically, the paper answers three key questions: How did the learning opportunities at COP24 matter? How were youth supported by their communities, and in turn how did they contribute back to their communities? How did their participation in these climate action activities contribute to youth leadership? By sharing this work, we aim to create an additional forum for youth voice on climate change as well as advance understanding of the value of Indigenous youth engagement in climate governance.

### 1.1. Literature Review

#### 1.1.1. Youth Well-Being

A growing number of studies have explored the impacts of climate change on the well-being of northern Indigenous peoples [13]. Climate-induced stress and threats to well-being on Indigenous youth have been of particular interest in recent years, although this work has been limited to the Inuit youth in the Arctic [11]. Social relationships and connections within the community, as well as participation in cultural events, traditions, and knowledge, have been identified as factors that contribute to youth's ability to cope with stress and improve their well-being [18]. This paper explores how climate action can be a landscape for youth to engage with their networks and culture, which in turn may have positive benefits to their well-being.

#### 1.1.2. Learning Outcomes—Climate Change

Climate change is a complex issue with many dimensions that affect the well-being of youth. In arctic communities, opportunities to learn about both the science and Indigenous knowledge related to climate impacts has been relatively limited. At the same time, public awareness about the "threat" and "stresses" of climate change has increased exponentially. The disconnect between knowledge and tools to cope and adapt as well as increasing "worry" about climate futures has seemingly led to increases in anxiety and stress among youth.

Climate related education is considered a useful strategy for addressing hopelessness and motivating action. At present, it is suggested that youth have a relatively limited understanding of both the causes and potential adaptive strategies available to address climate change [19]. However, simply sharing "information" about climate change is not the only solution, particularly for Indigenous youth. Research on climate change education, for example, demonstrates the critical importance of teaching and learning opportunities that weave together an increased understanding of key issues of climate change with narratives of hope and opportunity [10]. Culturally appropriate, place-based education and exposure to new opportunities for cross-cultural learning are among the strategies considered wise practice for Indigenous youth [20–22]. Collaborative learning, where students problem solve together, is another important approach and can lead to more cooperation, higher academic achievement, and greater self-esteem [23,24].

#### 1.1.3. Community Supports and Networks

Previous research has shown how individuals including youth can overcome stress, trauma, and other life challenges and build resilience by drawing from the social and cultural networks and practices that constitute communities [25]. This is particularly true in many remote and Indigenous communities where strong kinship networks and social connections have been the basis for resilience to other kinds of economic, health, and environmental stresses (e.g., variability in wildlife valued as food).

Family can be a particularly important source of strength and capacity to learn and cope with stressors, including those associated with climate change. "Family protective factors may increase prosocial behavior and resistance to the negative effects of crises or stress by providing a stable yet

flexible and supportive environment" [25]. Just as youth and their individual resilience may be informed and shaped by families, so too are the family unit and its behavior in the face of stress shaped by the cultural norms and practices of its community and broader society. In this way, individual social relations are interconnected with culture and identity-making, which is particularly important during adolescent years. Research with Sami youth, for example, has shown that: "the key to relational resilience seems to be enculturation—that is, the degree to which youth are embedded in Sami and local cultural traditions and in the practice of cultural values and ceremonies" [26]. Such enculturation and identity making is not only key to cultural continuity, but has tremendously important implications for youth well-being. For example, youth with strong social networks, including continuity of social and cultural networks, are characterized by lower rates of suicide and other kinds of social illness [4,26].

Such research is synergistic with theories on social capital which assert that "shared norms, values, beliefs, trust, networks, social relations, and institutions are important to collective action and other benefits" [27]. The influence of community and social capital on individual knowledge and behavior is not unidirectional. Like a bank, individuals can contribute to, or draw from, their social capital at various stages of life and in different contexts. Access to social capital and networks has been identified as an important way to "overcome the odds" and improve outcomes for marginalized youth, such as young Indigenous people [28]. While key research has focused on the importance of family and community to the resilience of youth, few studies have focused on the contributions of youth, including Indigenous youth, to the social capital and resilience of their communities to climate change. Our research explores the ways in which youth engagement in the COP24 meetings are supported by, but also contribute to, their families and communities.

### 1.1.4. Leadership

Little is understood about the role of Indigenous youth in northern governance; in other research it is often assumed that youth have little to contribute in policy spaces [29]. In recent years, stereotypes of youth apathy towards government have also increased. However, deeper study of youth motivations to engage in politics reveals a different kind of narrative. Although there is widespread skepticism about formal political parties, distrust in political figures, and a general sense of alienation from mainstream politics, much research suggests young people have a strong interest in "cause-oriented" political and social movements [30,31]. Leadership skills such as organization, public speaking, negotiation, and meeting participation are essential for those that want to take part in community change efforts [32]. The climate action movement has created new ways of thinking about the power and voice among youth, including Indigenous youth. Through this research, the aim has been to understand more about how youth perspectives about climate change and its impacts on their lives, communities, and environments can contribute to better decision-making. Conversely, how can participation in climate action activities lead to the development of youth leadership skills?

## 2. Materials and Methods

### 2.1. Setting

Youth from the Mackenzie River Basin attended the COP24 meetings in Katowice, Poland. Their participation was an extension of a Youth Knowledge Fair knowledge mobilization and capacity building activity, funded through Tracking Change. At this biennial event, 40 youth (Grades 10–11) from the Mackenzie River Basin shared ideas about climate change and learned from other youth, elders, graduate students, and faculty from the University of Alberta. The intent was to create opportunities for youth to learn from their own community members and other sources of knowledge about the impacts of climate change in their regions. Posters were created by each student, who then shared their work at the University of Alberta. Similar to a conventional "science fair", the intent was to showcase youth talents, creativity, and innovation. However, given the strong focus of the Tracking Change project on Indigenous Knowledge, the knowledge fair was unique in celebrating the value of

cultural worldviews, values, and knowledge systems of the Dene, Cree, and Inuvialuit students who participated. During the fall of 2018, six students who had participated in the knowledge fair were invited to share their work and participate in the COP24 meetings in Katowice, Poland. They attended and presented their work to the Canadian Council of the United Nations Educational, Scientific, and Cultural Organizations (UNESCO) in Paris, at the Conference of Youth (COY), the youth component of the Conference of Parties (COP) 24 in Katowice, Poland, as well as at Indigenous caucus meetings and media outlets. A key question following both the knowledge fair and the travel to Paris and Katowice was, how did these events benefit the youth involved?

The Youth Knowledge Fair (YKF) and trip to Europe are particularly interesting in the realm of youth development due to the challenging circumstances faced by some Indigenous communities. Intergenerational trauma, a result of colonialism and residential school, has contributed to high rates of Indigenous suicide [4,33,34], substance use disorders [3,35], and incarceration [36]. Due to the seriousness of these issues there are growing opportunities to engage with youth and improve their outcomes [37]. The Tracking Change events responded to community interest in having more youth engagement opportunities. The YKF was inspired by guidance from elders and leaders of the Tracking Change project who, concerned about limited educational opportunities for their children and grandchildren, emphasized that the project, "should do something for the youth".

*2.2. Procedures*

To answer these questions, fourteen (14) semi-directed interviews were completed with youth (4), coordinators (2), chaperones (4), and community members that were involved in some way in supporting youth (4). Interviewees were asked to participate in the study if they (a) were a participant at the Tracking Change Youth Knowledge Fair and trip to COP24, or (b) had a close connection with a participant. While the sample size of 14 is not a representative sample of Indigenous youth climate activists, the researchers were satisfied that enough interviews were completed to gain insight about the specific experiences and outcomes of the individuals that participated in these specific events. Two of the six youth that participated in the Tracking Change events decided not to participate in this research, although they did grant consent for other interviewees to reference their names publicly.

An interview guide was created based on the lead author's observations at the events, as well as through scoping discussions with organizers. Questions included: How did it feel to have the chance to share your story? Who supported you before, during, and after the trip? How do you think the trip impacted you? The interviews were carried out with careful consideration of ethics procedures for vulnerable persons, specifically youth under the age of consent (University of Alberta Research Ethics Board, Study ID: Pro00085621). Interviews were completed one year after the trip was completed, which allowed the research team time to secure resources, scope the project, and obtain ethics approval. A conventional content analysis approach [38,39] was employed to identify key themes and sub-themes in the transcripts (Table 1). Interviewees had 30 days to review and amend their transcript. This review period has been standard practice during previous research with Indigenous communities and was approved by the university ethics board. Participants reviewed drafts of this paper to ensure that the findings were reflective of their experience. Verifying data interpretation is an important part of completing research with Indigenous communities because research has been exploitative to their knowledge and resources [40,41]. The data is presented in a chronology that follows from the early period of youth planning for the YKF through to post-event reflections after the COP24 trip.

An emerging framework for understanding social capital as it relates to youth development is "webs of support". This model "actualizes how relationships and resources optimally operate to promote more accurate examinations of how adolescents gain the developmental supports necessary to thrive" [42] (p. 2). Through this exercise, we were able to visually depict the ways youth are supported by (and offer support back to) their communities, as well as map significant interactions and events throughout the Tracking Change events. Figures 1–4 were created by combining information from the semi-structured interviews with a mapping activity completed by the youth participants.

Diagrams created by youth were transposed into a digital format using Microsoft PowerPoint. A limitation of this method is that youth may not be engaged with the activity, which may mean the figures illustrated are not accurate. Supplementing the youth drawings with information from key informant interviews helped to address this limitation. The diagrams presented in this paper are simple and not intended to consider the difference in significance and frequency of youth relationships.

A limitation of this study is the small sample size (14). While interviewees were able to give a greater understanding of specific experiences and outcomes related to the Tracking Change events, these insights are not necessarily generalizable to other contexts due to the sample size. Another limitation is that the research was completed one year after the climate action activities, so it is limited in temporal scope. A longitudinal study would create insights into the benefits of activism in the long-term and where the youth end up as adults. More research on the value of youth engagement in this and similar forums would be useful in cementing understandings of the important and powerful role that Indigenous youth can have in addressing climate change and the broader benefits of their participation.

**Table 1.** List of key themes and subthemes identified through conventional content analysis.

| Key Themes | Subthemes |
| --- | --- |
| Challenges faced by youth | Limited opportunities |
| Youth leadership | Local; international |
| Networks created | Local; international |
| Trip impact | Immediate; long-term |
| Ways of learning | Land-based; Traditional Knowledge |
| Culture | Indigeneity |
| Temporal thinking | Ancestors; future generations |
| Ways of communicating | Storytelling; emotional |
| International experience | Culture shock; global thinking |
| Audience response | Actually listening; repeating message |

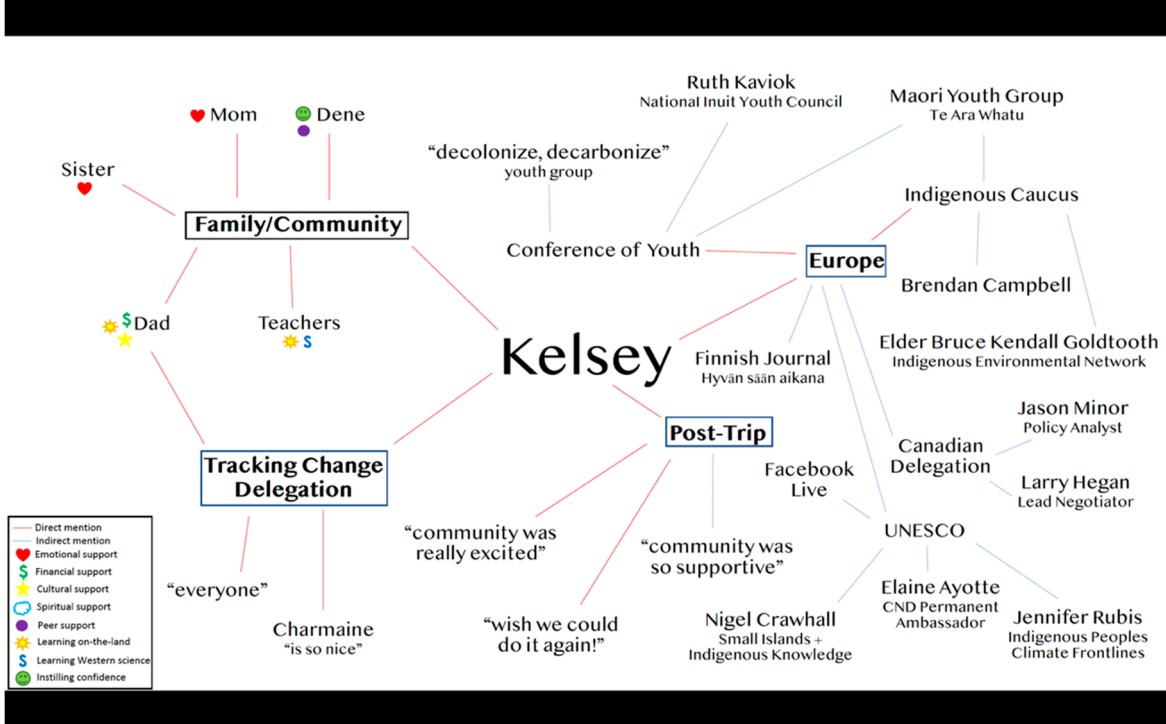

**Figure 1.** Kelsey's Web of Support.

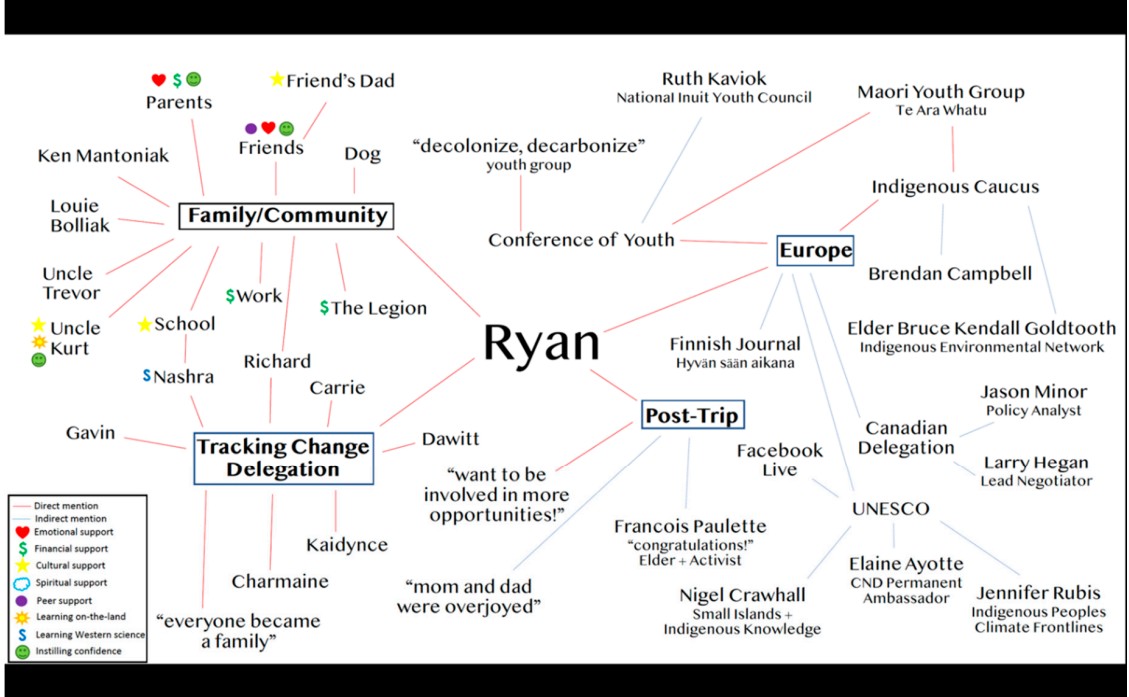

**Figure 2.** Ryan's Web of Support.

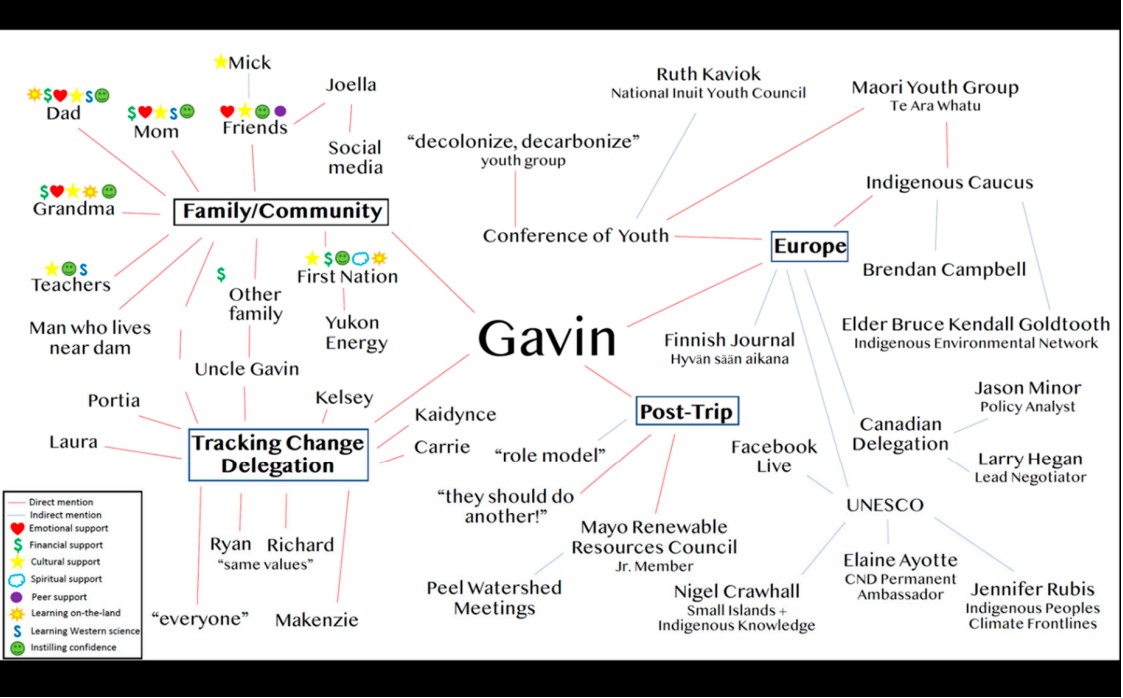

**Figure 3.** Gavin's Web of Support.

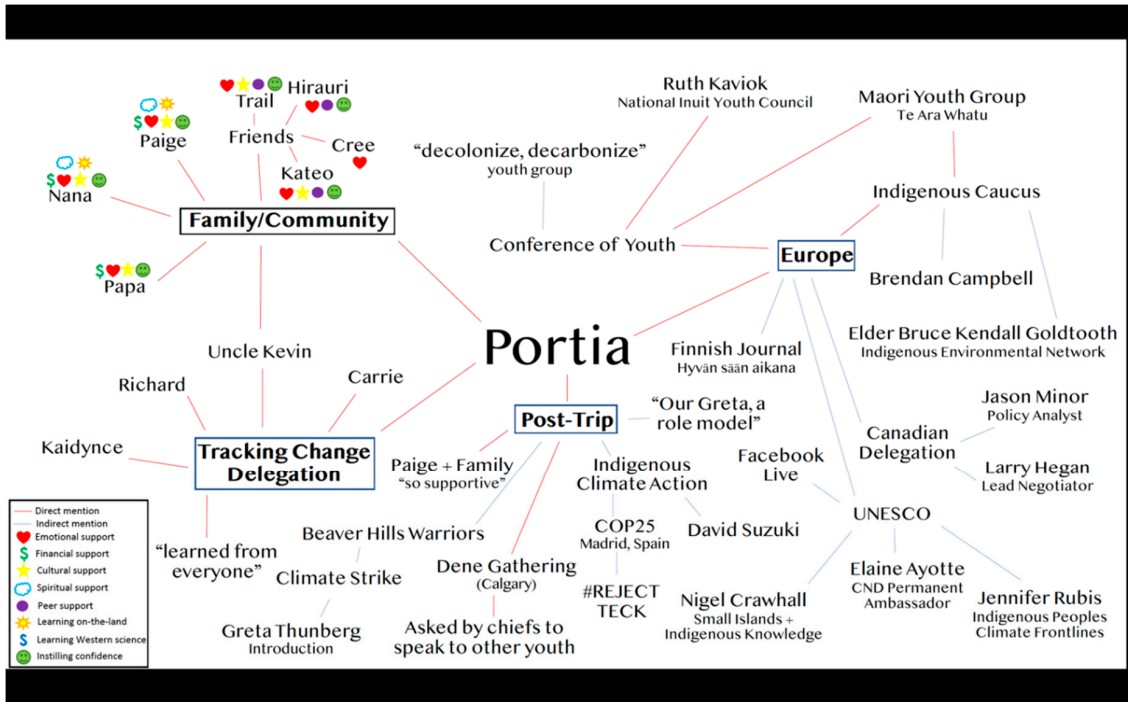

**Figure 4.** Portia's Web of Support.

## 3. Results

### 3.1. Learning Outcomes—Climate Change

Youth learning through the Tracking Change Youth Knowledge Fair involved recognizing and sharing the links between climate change and lived experience—learning that was embodied by and emotional for students and was connected to the real-life policy contexts of UNESCO and COP. As young people whose lives are deeply connected to the land, the students learned to recognize the impacts of climate change in their lived experience:

> *His Dad lives off the land. So he sees everything like permafrost melting, he sees the big sinkhole in the Dempster Road. He sees, like, how the climate is changing. He sees it because he lives it, he lives it.—he doesn't even need too much research because, I mean, he could see the impacts of climate change in his own life. Day to day.* (Dawitt, teacher/chaperone)

For one student, this was a new recognition that invigorated her sense of connection to the land:

> *Before, I didn't really care about the land, I guess, and that kind of sounds harsh but I didn't care. I didn't really know what was happening. I felt like learning about climate change—it really changed how important [the land] is to people and to respect the land.* (Kelsey Lockhart, youth participant)

This growing sense of connection to community and knowledge of the land was a key learning outcome for one community member, who saw the students' learning as a renewal of what had been removed through colonial history:

> *I think it's all about putting value on their connection to their community and their knowledge. And I think that there isn't enough value placed on young people being out fishing and connected to the land and giving fish to Elders. Like that has been taken away and diminished, but when people do those kinds of things they understand, like, the broader impacts.* (Joella Hogan, community member)

This learning was deeply emotional, motivating action and creating connections between the youth and their audiences as climate science "came to life" through the students' storied expressions of learning:

*I think Richard's stories about the land were very impactful, 'cause they really spoke from the heart; you could really almost feel his passion... [And Kaidynce told] the story of the belugas going inside the harbor when they normally never go due to climate change. I feel like the stories were so real and like they really showed a new perspective we're not used to. And I am sure there were a lot of youth [that] were listening to such an honest and truthful ways of displaying change. That I think everyone could, like, relate at a very deep level. To the stories.* (Laura Gaitan, chaperone)

While some expressions of emotion were hopeful, one youth expressed upset and frustration at her learnings that motivated her subsequent activism:

*It was kind of upsetting that climate change not only effects my reserve but, like, everywhere else. I've seen similar issues, like how the waters are being affected—how people are restricted from hunting because of the diseases that animals carry. And it was upsetting and was a real eye opener for me that made me want to pursue this activism.* (Paige Chisaakay, youth family member)

Community and intergenerational connectedness was key to processing these emotions, and a sense of relationality to one another within the group that travelled to Europe enabled everyone—not only the youth—to learn from one another. As one chaperone expresses:

*It was an emotional trip for sure. But it's funny, you're right. We did become a family. As a group with the chaperones, with the students... We all were learning so much from [each other].* (Nashra Kamal, teacher/chaperone)

A number of the youth and their chaperones emphasized how student learning was reinforced through a sense that they were actually listened to by their different audiences. Specifically, a few participants spoke of the uniqueness and potential impact of the students' experiential and land-based knowledge that is not "easily Googleable" (Nashra Kamal, teacher/chaperone). While the students' recognition by policy makers does not define the validity of their knowledge, this experience did reinforce their learning, including its unique and necessary position within a broader international dialogue.

*It was nice. People were actually listening ... I've told the story a couple of times [at home] in Mayo and no one listened. And then it's nice when people are listening and paying attention.* (Gavin Winter-Sinnott, youth participant)

*... it's kind of a wake-up call for the adults that may or may not think climate change is real ... So, to hear that from the youth is very eye opening.* (Kevin Ahkimnachie, family member/chaperone)

The cross-cultural, international nature of the trip provided an opportunity for new learnings of the world and the scope of its environmental problems. Conversations, storytelling, and attending presentations provided greater understanding of transnational, interconnected, and overlapping climate change impacts that expanded students' sense of the issue beyond—yet also in connection with—their local context.

*Well [meeting people] and getting to talk to them about the things that they face with climate change and how similar some of these [impacts] are to them. I met one lady, I forget where she was from, but she was telling me that she has exactly the same problems that we have here in Canada—it was so interesting getting to talk to these people about things that they experience and that I never knew happened around the world.* (Portia Morin, youth participant)

Watching the youth share their knowledge in this international context, chaperones emphasized a desire to build a strong sense of Indigenous knowledge of their changing lands in the young people.

*I think in a way we're not just sharing the knowledge, we're celebrating the knowledge that's being shared—giving it a platform to be shared amongst others, we would be lucky to be witness to it. It's so authentic and it's raw, it's real, I think it's very powerful . . . (Nashra Kamal, teacher/chaperone)*

At the same time, chaperones resisted placing the weight of responsibility to solve climate issues directly on the youth.

*You know also it was a bit of a burden on the youth because they [Indigenous caucus] were asking them questions like, "how can we help?" and I felt like . . . well those questions can be very useful or they can be very heavy loaded because youths don't necessarily have the answers, you know? We can't say "this is a policy change that needs to happen so that my community doesn't, like, sink in the ocean". (Laura Gaitan, chaperone)*

The focus for chaperones like Laura, therefore, was on the development and legitimation of land-based knowledge, rather than on solution-oriented climate learning.

*3.2. Family and Community Connections*

Community connections before and during the trip evidence the integrated nature of young people's learning and wellbeing—how family, community, care, culture, and learning are integrated for these young people.

Youth became involved in the knowledge fair through various connections in their communities including friends, high school teachers, leaders, and family members. Learning about the opportunity from someone close to them or a known and respected person in a position of trust was important.

*I was playing basketball in the gym and my mom told me that [our friend] Joella knew about this project to work on so I said yeah . . . Joella told my mom, my mom told me, and then I made a poster, went on the trip. (Gavin Winter-Sinnott, youth participant)*

The project was set up for equal participation by boys and girls, noted by one community member as an important strength and opportunity for young men to grow and develop.

*I really notice—in our community anyway—it always seems like it's girls that are getting more attention to travel and do these kinds of projects and I think it was nice to see a young guy get this . . . I think also like sometimes men's roles in hunting and fishing and on-the-land things is diminished. So, it's empowering young men to have a bigger voice and connection. (Joella Hogan, community member)*

The Youth Knowledge Fair was a unique opportunity for youth to build relationships and support one another. It required them to carry out research in their own communities. The learning environment of the Youth Knowledge Fair was characterized by a kind of cultural safety where youth felt comfortable to share their ideas. The opportunity for youth to get to know each other and "discover connections" was a noteworthy aspect of the learning process, intentionally created by the knowledge fair organizer to counter the disconnect that characterizes the colonial situation in Canada. In sharing their posters, the students could tell stories and point to photos of their aunties, Chiefs, community members, or sisters. In listening to one another's presentations, the students began to recognize how they were connected and how their lands were connected, and could identify with their knowledges.

One teacher commented that it was great to see the celebration of what youth know and the value of creating a platform for them to share their knowledge.

*I think that [Youth Knowledge Fair] was [an] excellent starting point. I remember Ryan saying one of the students that came was his cousin, like he was related to him, and I think that's the theme of the North, it's like a very family style outlook and I think that first event kind of represented that because there was a lot of representation from the North and I really like how it catered to Indigenous*

*students and [felt] like you were celebrating what they knew and you were giving them that platform to share their knowledge, right? So, in that respect I really liked that part of it. That setup was great, I think. And I think it helped to bring confidence for a lot of the students.* (Nashra Kamal, teacher/chaperone)

Community members rallied around the youth as they prepared to go on the trip to Europe. Beyond assisting with the creation of posters, they offered encouragement and support to the young people.

*They were extremely proud of him. I was proud of him . . . [it was not only me], but the whole community. We didn't know if there would be any financial support, so the community contributed. The whole community fundraised, they helped, they contributed. They were saying "he's a voice for us. He's a voice of our community".* (Dawitt, teacher/chaperone)

The connections between the youth and their chaperones were important. Chaperones were mentors and emotional supports to help youth navigate the stresses of travel and engaging in global events. As described by one participant, chaperones were also key to intergenerational knowledge sharing; young people could learn from the chaperones and problem solve together.

*I kind of remember [doing a bad presentation] when we were at UNESCO cause my poster wasn't really all together. But once I was with Dawitt he really helped me talk it all together . . . When we had to break into groups and work with other adults. I worked with Dawitt and he, like, really showed me how to put it together. I had all the facts there, I just had to figure out how to, like, word it out. So Dawitt really helped with that.* (Ryan Schaefer, youth participant)

Youth built relationships and connections with other Indigenous delegates, especially with a prominent activist by the name of Elder Bruce Kendall Goldtooth and a Maori group of young people from New Zealand, Te Ara Whatu. They were inspired, listened to, and validated by these delegates through expressions of transnational solidarity. For instance, Te Ara Whatu repeatedly reinforced the young people's learning and honored their presentations with a song at the UN Conference of Youth. One participant responded regarding the immediate and long-term impacts of this connection:

*I think [I was the most inspired by] the Maori group. I loved them. And I'm still kind of, like, you know, friends with them. Sometimes, here and there, I talk to them. I was so inspired by them and how they would talk to us. And I loved it when they, when we, went to this one place in Katowice and they sang to us there. I loved that . . . it was so beautiful.* (Portia Morin, youth participant)

Many of these connections were not superficial, but were raw and emotional.

*The elder he came, and he shook [the student's hand] . . . and then he said "I know this hand, and where it came from. It came right from the land." You know it was rough, it was, you know, like a person who lives off the land, and he said, like, he was, like, touching, like, his own hand.* (Dawitt, teacher/chaperone)

*I think that one [Indigenous caucus meeting] was very meaningful because it was, like—they were all Indigenous, so they got it, and they [understood] the pain. They were very receptive to, like, the pain that the youth were displaying. Like at UNESCO, it was interesting, but people were, like, "woah woah", like that is so cool "youth talking" they were, like, listening in that [Indigenous caucus] meeting, we could go on a deeper level and they were, like, "oh yeah. Like, this is happening. It is really devastating. It's really sad. Like we are losing our culture, we are losing our knowledge" and the Indigenous delegates can really relate.* (Laura Gaitan, chaperone)

An important outcome of these new relationships and connections was a growing awareness of climate change as a global issue.

*I thought it was really cool to see . . . kids around the world presenting their problems with climate change . . . But I was thinking "holy man, people actually care about this". I see it on the news and it's like nobody really else cares about it but when you go to all these giant conferences and people actually go and talk about it and stuff, and it really surprised me to see how many people actually do care about it.* (Ryan Schaefer, youth participant)

Multiple interviewees described the importance to community members of seeing local issues shared around the world, along with the continued sharing back at home over social media and CBC radio. Connectivity of social media and conventional press coverage allowed for networking across borders.

*I was sharing lots of things, like, through the Tracking Change posts that were going on—I was always sharing those in the community, and at the time I was also doing the First Nations social media so then I'd re-share from, like, the First Nation page too; that way more people saw it.* (Joella Hogan, community member)

*[When I saw the social media posts] I just thought it was amazing that she had that opportunity to go to Poland, speak about our community, speak about climate change, environmental issues. And she's so young. I was really proud of her. And so was our family. And our community. And I'm just glad that she got that opportunity to go speak about issues that we face.* (Paige Chisaakay, youth family member)

*The [community's response] was big . . . everybody was congratulating him and even me. [It was about] the awareness that he brought. He really didn't know he would have that impact, but, I mean, the awareness he brought [was important]. Like it was the talk of his hometown in Inuvik and even on CBC [radio] . . . and in Yellowknife again after the trip. So, it was, like, huge. The elders in the community they were saying "we're very proud of him".* (Dawitt, teacher/chaperone)

### 3.3. Leadership

Exclusion of Indigenous peoples and youth in policy processes, as well as the value of their perspectives, was highlighted as a key reason for increasing leadership and involvement.

*Well I think Indigenous peoples are really pushed to the side in most cases, so we don't have a [chance to] speak; when we do have the chance to speak we're kind of, kind of, put down for it . . . . I think that's why it's important that Indigenous youth need to be heard, need to gain more knowledge on this.* (Paige Chisaakay, youth family member)

*. . . not a lot of people take time to realize that youth are powerful with the way we use our minds. We're powerful with the way we contribute and receive our knowledge. So, yeah, I think that's why a lot of youth need to be more involved. They need to—even if you don't know or think they know a lot of things, you know?* (Portia Morin, youth participant)

An important opportunity and outcome of the participation in the COP24 meetings, as well as in the knowledge fair in Edmonton, was confidence building which, as described by a chaperone, comes from learning to trust oneself.

*I think Ryan learned to trust himself. He kind of gained confidence as he went along. Like before he was—like, in a panicked state. He was very nervous about having to present and trying [to] meet people that have power and making deals in the world, right? He was very apprehensive about it and I had to just go along and support him. [He also got support from] the people that we presented to—like, complete strangers. I think, as they went along, he just became more confident with what he was saying, more sure of himself. I think this is an excellent confidence building opportunity for him.* (Nashra Kamal, teacher/chaperone)

In keeping with the sense of connectedness described above, youth supported and inspired one another. The Tracking Change trip peer group created its own kind of internal confidence.

*I wasn't the first to do the presentation so when I saw the other kids doing it I thought, I gotta be able to do it too then.* (Gavin Winter-Sinnott, youth participant)

The cultural safety and the social support surrounding the Youth Knowledge Fair and Europe trip seemed to translate into confidence in speaking and talking about issues of importance, as reiterated by many interviewees. As said by one teacher, it was almost like the student "released the lion within":

*I was flabbergasted . . . The passion, the seriousness—oh my goodness. I know Richard was a little bit passionate but when he saw Aboriginal kids from other communities and their passion, it was kind of like the "lion from within" [comes out]. Like, I mean, before they really didn't want to do that much, and we didn't know [how it would go] at the beginning . . . But when they saw all the kids working and the issues, they were like really, really very encouraged. And they were, like, really passionate.* (Dawitt, teacher/chaperone)

This confidence translated to material impacts for one youth, who presented his poster in his community and saw a real-world response before leaving for Europe. He describes how his local energy board took up his poster's argument about water levels and released water from the local dam.

*Yukon Energy put on a public meeting to talk about what was going on at the lake. And then I brought my poster and told them where I was bringing the poster [to Europe] to talk about [the water levels]. And then a couple weeks after that the water levels started to rise up and rise up, rise up.* (Gavin Winter-Sinnott, youth participant)

Students and chaperones were surprised and inspired by the confidence, eloquence, and passion that the youth developed on the trip. Their strong leadership has translated to continuing learning, activism, and involvement in the environmental field. For instance, Portia became involved in climate activism, and Gavin is sitting on his community's Renewable Resources Council:

*There was a huge Peel Watershed meeting in the community and Gavin came and I think he even asked a question, which is good because not very many young people go to those kinds of meetings. He also stepped up to [work] on the Renewable Resources Council [which] was looking for young people. Because he's connected to the land and the people and he knows stuff he cares about what's going on, and just to have a youth voice on there. . . . but you have to be confident, and I don't think that many young people have [that confidence], but he did—he's not that shy to speak up when it's something important to him.* (Joella Hogan, community member)

*. . . now he's studying renewable resources and that's pretty exciting too. I'm pretty sure he's probably the only one in his program that's attended any kind of event or project like that. And that's kind of significant. Like someone joked in our community, "he should be teaching those classes" right? Just to have that kind of experience at his age already is pretty significant.* (Joella Hogan, community member)

The youth's leadership in climate action inspires other young people to get involved in their communities and in causes they are passionate about.

*. . . they're creating that ripple effect . . . [we need] youth champions in communities. But how do we support them? How do we bring them out to be to be heard more?* (Sharlene Alook, community member)

*. . . there are lots of youth who are inspired because of him. And they were even asking me if there is [going to be another event like this] in future—they want to be a part of it.* (Dawitt, teacher/chaperone)

Interviewees noted the significant challenges that youth are inheriting. One grandmother shared a story which emphasized the importance of culture in the youth's ability to take leadership roles in addressing issues.

> *[The prophet] was saying that, when there are forest fires and deforestation in the boreal forest of northern Alberta, people [will be] scared and afraid and they won't know where to go. The road [will be] cut off and [people] won't be able to see where they are . . . And you have to talk to the young people—he [the prophet] said that the young people have to hear our language, our stories, and to know who they are and where they are going.* (Molly Chisaakay, Elder/family member)

*3.4. Webs of Support*

The final part of the results section provides a stylized summary of the webs of support associated with the youth that participated in the Tracking Change events. Figures 1–4 are digitized versions of a mapping activity that the four youth participants conducted as part of their interviews. The red lines and symbols are actors that the youth explicitly identified as important to them, whereas the blue lines are actors that key informants stated were important to the youth. Thus, these figures are a combination of hand drawn maps by the youth participants and the feedback and input of their loved ones. Including observations from adult informants supplemented the youth interviews and provided greater insight about networks related to the trip. The four nodes indicate key periods (Europe and post-trip) and key communities (the youth's families/communities and the Tracking Change delegation) in the students' experiences. These nodes were selected because they create insights on important actors and events as related to the youth's activism. Webs of support are presented for the four youth participants because this paper is focused on understanding the importance of networks in their development.

There are several key takeaways from these webs of support. First, each web of support illustrates the significance of family and friends. The concentration of colorful symbols in each family/community network indicates the importance of each actor to the young people. Parents and friends were identified as important sources of support and knowledge about the land. Western ways of learning (i.e., schools) are less important to the young people's understandings of the environment. These local actors were critical in giving youth the knowledge and emotional, cultural, and material support to take on the challenge of being activists.

While the Tracking Change delegation node was temporary in nature (lasting approximately two weeks) it is relevant that each youth indicated that "everyone" on the trip was a source of support during the time in Europe. Ryan (Figure 2) remarked that "everyone became a family", and Gavin (Figure 3) said he felt connected to other youth on the trip because they had the "same values" and similar on-the-land experiences. Chaperones were discussed as a useful practical support when it came to preparing the youth's research posters and presentations. The youth mentioned that, when they were feeling nervous about presenting, they leaned on each other; seeing their fellow delegates sharing their stories empowered them to present publicly as well. This aspect of the webs of support may indicate that the positive student-student, student-chaperone relationships were important for the young people's participation in Europe.

One particularly interesting insight is related to the Europe node where the students and interviewees discussed important actors and events for the young people. Each youth participant emphasized the significance of meeting other Indigenous activists at the Indigenous Caucus meeting and Conference of Youth, as well as of participating in Indigenous-centric policy spaces. Meetings involving Government of Canada delegates, western media, and UNESCO representatives were highlighted as important by other key informants, but the youth themselves did not feel particularly connected to those spaces. This insight seems to indicate that youth may place importance on a shared Indigenous identity and connection with other Indigenous peoples. Many of the Indigenous leaders also took the time to mentor and encourage students, listen respectfully, and engage knowledge. Perhaps these people and spaces were more comfortable and familiar to the young people given that

they were in such a foreign environment (e.g., in a new country; participating in large-scale events). It could also be that youth were empowered by solidarity with other Indigenous peoples while sharing knowledge in western policy spaces. The figures also illustrate the breadth of connections the students were able to make. Through social media, their messages extended beyond geographic borders and the confines of meeting rooms. Youth were able to share their local experiences of climate change with people from different parts of the world.

Finally, the post-trip node illustrates how the youth were received by their communities after participation; it also provides a snapshot of their continued activism. Figures 2 and 3 illustrate that youth themselves felt very much supported by their communities when they returned from their trips. Key informant interviews supplemented the youth's comments by discussing local responses to the youth's return to the community. In Ryan's case, Francois Paulette, a well-respected Elder and conservationist, congratulated and thanked him for his activism (Figure 2). Chaperones remarked that the students were viewed as role models for other youth in the community upon their return. Gavin's web of support (Figure 3) shows that he continued to be involved in environmental governance by serving as a junior member on the Mayo Renewable Resources Council. A key informant explained that, as part of that role, Gavin attended Peel Watershed meetings where he talked about his experiences in Europe. Figure 4 illustrates Portia's ongoing, passionate activism. Since participating in the Tracking Change events, she has continued to be involved in movements related to youth leadership in Indigenous communities and anti-fossil fuel campaigns, as well as speaking at Climate Strikes and attending COP25. When asked if the trip to Europe had contributed to her ongoing activism she replied, "yes, yes, yes, yes, yes!" All of the youth participants interviewed explained that they felt they had a greater understanding of climate change after participation, and every research participant stated that the leadership skills the students developed through their activism may be useful if they continue to be involved in issues that matter to them.

## 4. Discussion and Conclusions

Warming temperatures, extreme weather events, and other climate related changes are impacting on northern communities in Canada and elsewhere [1]. The lived experience of these changes, coupled with discourse from education and media sources, may be contributing to anxiety and hopelessness among youth, including those living of the Mackenzie River Basin [7–10]. Research elsewhere suggests that a key pathway to offsetting such climate anxiety and building hope and resilience may be through youth participation in climate governance [11]. However, little of this work has engaged Indigenous youth. In this paper, we have addressed this gap by sharing the narratives of four youth (and their families and related community members) who attended the 2018 Conference of the Parties on Climate Change in Katowice, Poland (COP24) in 2018.

We have specifically addressed the often used stereotype that youth have little to say or are apathetic about politics, including that which concerns climate change [29]. The interview outcomes evidence a breadth of ideas, concern, and leadership values as well as interest in engaging in governance. Specifically, there were expressions of pride about culture, care for the land, and their communities. The experience of participating in COP24 engendered personal growth and confidence to share their feelings, experiences, and ideas about climate change and its impacts on their communities. The narratives shared in this paper also evidence markers of hope and efficacy about "being heard" by other Indigenous peoples in other regions and those in power in their own communities and regions.

The work also resonates with theories on social capital. The social supports that the youth mapped (Figures 1–4) detail the important role of families, friends, and communities as well as the roles of individuals met through the COP24 event. We suggest that expanding social networks can enable youth, particularly from remote communities, to tap into new opportunities which they may be otherwise unable to access in their own communities. This is meaningful in a variety of ways. As noted by previous scholars, youth who are able to "overcome the odds" usually do so through networks that connect the young people to new resources or opportunities [28]. Other studies have found that

collaborative learning opportunities are a way for students to develop social skills for cooperation and conflict resolution [23] and improve self-esteem [24]. However, given colonial histories and the impacts of outsider education of Indigenous youth (e.g., residential school systems), it is recognized that the expansion of global teaching and learning opportunities cannot come at the cost of locally generated, family centered, place-based, and culturally constructed opportunities.

In summary, the paper has shared the voices of youth as well as chaperones, teachers, and community members who supported youth at the COP24 meeting in Katowice, Poland. Based on the interviewee narratives, it seems that the combined learning opportunities, support of the community, and newly developed leadership skills created through the events had an impact on all those who participated. The benefits of Indigenous youth activism are not limited to the participants but may also have cascading positive effects in their northern communities as well as on those who shared in the youth experience of COP24 in 2018.

**Author Contributions:** Conceptualization, M.M., B.P., and C.K.; methodology, M.M. and B.P.; validation, M.M., B.P., and C.K.; formal analysis, M.M.; investigation, M.M.; resources, B.P.; writing—original draft preparation, M.M.; writing—review and editing, B.P.; project administration, M.M.; funding acquisition, B.P. All authors have read and agreed to the published version of the manuscript.

**Funding:** The research was funded by a grant from the Social Sciences and Humanities Research Council (www.trackingchange.ca) and through the support of the Faculty of Agricultural, Life, and Environmental Sciences, University of Alberta Canada.

**Acknowledgments:** We thank Paige Chisakaay, Kelsey Lockhart, Portia Morin, Ryan Schaefer, and Gavin Winter-Sinnott for their enthusiastic involvement in the Tracking Change events and for providing comments on this paper.

**Conflicts of Interest:** The authors declare no conflict of interest.

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
