# Peer review of "Youth Engagement in Climate Change Action: Case Study on Indigenous Youth at COP24"

_sustainability, doi:10.3390/su12166299_

Round 1

Reviewer 1 Report

The paper addresses the pressing issue of youth engagement in the climate change action, offering the interesting angle of analysing indigenous youth in Alberta, Canada.

The methodology is sound, although a sample limited to 14 interviews (of which only 4 youth) only may not be representative. It would be useful to know how you selected the interviewees, based on which criteria. Also, it would be god to show the coding (key themes and sub themes).

The result section is very long and with too many direct quotations from the interviews. In the first part of this heading, a general explanation and description of the study areas have been provided without a deep interpretation in this section. Further interpretation is essential in the conclusion and discussion sections, where the reader is also keen to know whether or not the results of this paper can be applicable to the other contexts.

A final small remark: a ‘proper’ literature review heading is missing. You just need to split the long Introduction into a shorter one, in which you highlight the background and aims of the research and state your questions, and a section with the rest of the introduction, which can be part of your literature review. As you aim to contribute to the literature on youth well-being, you may want to add a couple of paragraph to extend your literature here.

Author Response

The paper addresses the pressing issue of youth engagement in the climate change action, offering the interesting angle of analysing indigenous youth in Alberta, Canada.

The methodology is sound, although a sample limited to 14 interviews (of which only 4 youth) only may not be representative. It would be useful to know how you selected the interviewees, based on which criteria. Also, it would be god to show the coding (key themes and sub themes).

Thank you for taking the time to review. We have added to the methods in response.

The result section is very long and with too many direct quotations from the interviews.

Thanks. We have cut down the results section to only include quotes and discussion related to literature on youth well-being, climate change learning, leadership, and networks.

In the first part of this heading, a general explanation and description of the study areas have been provided without a deep interpretation in this section. Further interpretation is essential in the conclusion and discussion sections, where the reader is also keen to know whether or not the results of this paper can be applicable to the other contexts.

We have added 14 references to better explain the value of our research to the literature on youth well-being, climate change learning, leadership, and networks. The discussion has been refined to make these connections clearer. We have also removed unsubstantiated statements from the paper to better explain the value of the research to the context of literature on Indigenous youth in climate action.

A final small remark: a ‘proper’ literature review heading is missing. You just need to split the long Introduction into a shorter one, in which you highlight the background and aims of the research and state your questions, and a section with the rest of the introduction, which can be part of your literature review. As you aim to contribute to the literature on youth well-being, you may want to add a couple of paragraph to extend your literature here.

We have restructured the introduction to include formal lit review heading. We have also added a paragraph on Indigenous youth well-being.

Reviewer 2 Report

The climate change and the response to it by the youth is a very important issue, but this paper fails to advance the knowledge in the area. The methodology is full of bias and over strong sense of authors' confirmation bias is leading the manuscript. 

Perhaps this could be transformed to a commentary, essay?

Below more detailed comments:

Abstract: It doesn't clearly state what the outcome of the research was. Moreover, there is a reference to "this project", but no description of the project was preceding.

Methodology: The semi-directed interviews are reported as a scientific method to answer the research questions about the benefits for the youth from activism. However, the sample size consisting of only 4 youth representatives seems small and would not convincingly support the results. At the same time, authors report there were 6 young individuals going for event in Europe, it remains unclear why the 2 people didn't take part in the interview - an attrition bias. 

Authors should attempt to justify small sample size sufficiency in the light of general recommendations for the qualitative content analysis to be between 20a and 30 interviews.  

What were the socio-demographic data of participants?

What was the timing of the recorded interviews? (how many days after the trip?)

What is the rationale for giving participants 30 days to modify the recorded interview data? It is a potential source of biases. Also, how the modifications into recorded footage were administered by participants? 

What do authors mean by "the data was verified with participants to ensure that findings were reflective of their experience"? This seems like the researcher could have too much direct influence on the responses, through double-checking if they are "correct". It looks like a serious shortcoming in the methodology and another source of bias. 

[line 200] limitations should be listed at the end of the manuscript, in the discussion section.

Results: The Results are not convincing due to the flawed methodology. From the interview transcripts, it seems that children were just excited about the trip, so it could be about virtually anything. The rest was directed by adults...

The reported positive psychological effects of the trip on kids could be due to just international exposure, broadening the horizon through travel, not climate change activism. 

Ethics concern: Why are the personal particulars of the youth provided in the manuscript? As they're part of vulnerable groups, it shouldn't be the case.

Author Response

The climate change and the response to it by the youth is a very important issue, but this paper fails to advance the knowledge in the area. The methodology is full of bias and over strong sense of authors' confirmation bias is leading the manuscript. 

Perhaps this could be transformed to a commentary, essay?

Thank you for taking the time to review. We have made revisions to the paper which clarify the methodology, acknowledge limitations, and illustrate the value of the work in the context of literature on youth well-being, climate change learning, leadership, and networks.

Below more detailed comments:

Abstract: It doesn't clearly state what the outcome of the research was. Moreover, there is a reference to "this project", but no description of the project was preceding.

Thanks. We have revised the abstract for clarity.

Methodology: The semi-directed interviews are reported as a scientific method to answer the research questions about the benefits for the youth from activism. However, the sample size consisting of only 4 youth representatives seems small and would not convincingly support the results. At the same time, authors report there were 6 young individuals going for event in Europe, it remains unclear why the 2 people didn't take part in the interview - an attrition bias. 

Thank you for raising this important point. We have added to the methods to acknowledge the limited sample size.

Authors should attempt to justify small sample size sufficiency in the light of general recommendations for the qualitative content analysis to be between 20a and 30 interviews.  

We have added to the methods to acknowledge the limited sample size and explain why we were satisfied with 14 participants.

What were the socio-demographic data of participants?

Thanks for your comment. Specific socio-demographic data of individual participants is not relevant to this study.

We have expanded on our literature review to provide greater context on the socio-economic conditions faced by many Indigenous youth in Canada’s north. We provide context of some of the challenges facing Indigenous communities as a result of residential schools and ongoing colonization, and how these stresses are relevant when doing research with young Indigenous peoples.

What was the timing of the recorded interviews? (how many days after the trip?)

We have added this to the methods.

Interviews were completed one year after the trip which allowed the research team time to secure resources, scope the project, and receive ethics approval. We acknowledge this as a potential limitation in the conclusion.

What is the rationale for giving participants 30 days to modify the recorded interview data? It is a potential source of biases.

Thanks. We have added this to the methods.

Also, how the modifications into recorded footage were administered by participants? 

Thanks. We are not sure what this comment is referring to.

Youth completed the mapping activity by drawing on paper and the lead author then digitized these drawings using PowerPoint.

What do authors mean by "the data was verified with participants to ensure that findings were reflective of their experience"? This seems like the researcher could have too much direct influence on the responses, through double-checking if they are "correct". It looks like a serious shortcoming in the methodology and another source of bias. 

Thanks for this comment. We have further explained this in the methods section and provided 2 sources to support use of verification (Fletcher, 2013; Smith, 2013).

[line 200] limitations should be listed at the end of the manuscript, in the discussion section.

We have added two limitations to the conclusion: the limited sample size (not representative of entire population) and the time period that data was collected (study completed one year after the trip does not prove there are long-term, lasting benefits of participation).  

Results: The Results are not convincing due to the flawed methodology. From the interview transcripts, it seems that children were just excited about the trip, so it could be about virtually anything. The rest was directed by adults...

The reported positive psychological effects of the trip on kids could be due to just international exposure, broadening the horizon through travel, not climate change activism. 

Thank you for raising this point.

We have added 14 references to better explain the value of our research to the literature on youth well-being, climate change learning, leadership, and networks. The discussion has been refined to make these connections clearer. We have also removed unsubstantiated statements from the paper to better explain the value of the research to the context of literature on Indigenous youth in climate action.

Ethics concern: Why are the personal particulars of the youth provided in the manuscript? As they're part of vulnerable groups, it shouldn't be the case.

Thanks. The focus of the Tracking Change events was to create space for and elevate youth voices, so we gave youth (as well as adults) the opportunity to choose to include or exclude their names. In the event the youth was a minority we sought “assent” from the young person and “consent” from their parent/guardian. This procedure was agreed upon by the research ethics board. 

Additionally, the highly publicized nature of the trip meant that interview participants may be identified even if they were assigned a pseudonym. This potential risk was explained to participants before they completed an interview.

Reviewer 3 Report

Dear Author(s),

The topic developed is of interest and relevance. Your manuscript –in my opinion– offers an interesting contribution in the sustainability context. The commitment of youth is critical, in general, and especially in the fight against climate change. This topic could provide an important contribution to this research area.

Author Response

The topic developed is of interest and relevance. Your manuscript –in my opinion– offers an interesting contribution in the sustainability context. The commitment of youth is critical, in general, and especially in the fight against climate change. This topic could provide an important contribution to this research area.

Thank you for taking the time to review.

Round 2

Reviewer 2 Report

Regardless of the substantial amount of work that authors put into improving this version of the manuscript, the work still fails to convincingly advance the knowledge in the area of indigenous communities and climate change activism. Reported findings are very likely due to random sampling variability and there is a very low level of confidence in that they reflect real phenomena.

After the round of review,  more very serious limitations have been uncovered such as conducting interviews one year after the event. This and small sample size (not 14, but 4 subjects representing the youth), are the main reasons for rejection, and these, apparently cannot be resolved without conducting a new study. 

The research work on topics of that importance has to be conducted with extra care to methodology, transparency and objectivity, otherwise, the pressing issue will not be considered seriously by anyone. 

I maintain that this work could provide interesting content for a commentary, essay, but not a scientific journal article.

Author Response

Regardless of the substantial amount of work that authors put into improving this version of the manuscript, the work still fails to convincingly advance the knowledge in the area of indigenous communities and climate change activism. Reported findings are very likely due to random sampling variability and there is a very low level of confidence in that they reflect real phenomena.

While there are a growing number of papers related to climate activism, few of these papers deal with youth experiences. Furthermore, there are currently no other papers in the literature that share the voices of Indigenous youth about a specific climate action activity such as COP24. We respectfully disagree with Reviewer 2 that the paper does not make a convincing argument about its contribution to the climate action literature. We would invite the reviewer to consider whether his/her concerns stem from a poor understanding of the literature related to Indigenous peoples in Canada and/or a narrow and (seemingly) white / elitist view of climate politics. There are no other papers in the literature that bring forth the voices of this demographic; as authors we are concerned that the potential dismissal of this paper reflects poorly on the openness of scholars in this field of inquiry to the voices of Indigenous peoples.

After the round of review,  more very serious limitations have been uncovered such as conducting interviews one year after the event. This and small sample size (not 14, but 4 subjects representing the youth), are the main reasons for rejection, and these, apparently cannot be resolved without conducting a new study. 

Regarding concerns that the interviews were conducted one year after the event, we would point to myriad studies that engage interviewees after the fact.  This ranges from harvest, food and nutrition studies to work on climate action. For example, Fisher (2016) wrote a paper on youth recollections of their early life influences in climate action. On that basis, the reviewer's concern that interviews were done up to a year after the events is a poor rationale for rejecting this manuscript. We would invite the reviewer to expand their knowledge of the breadth of social science research methods including recall studies. --- Fisher, S. R. (2016). Life trajectories of youth committing to climate activism. Environmental Education Research, 22(2), 229-247.

The research work on topics of that importance has to be conducted with extra care to methodology, transparency and objectivity, otherwise, the pressing issue will not be considered seriously by anyone. 

I maintain that this work could provide interesting content for a commentary, essay, but not a scientific journal article.